# Design of mmWave Directional Antenna for Enhanced 5G Broadcasting Coverage

**DOI:** 10.3390/s21030746

**Published:** 2021-01-22

**Authors:** Tao Hong, Shuli Zheng, Rongke Liu, Weiting Zhao

**Affiliations:** 1Yunnan Innovation Institute·BUAA, Kunming 650000, China; 2School of Electronics and Information Engineering, Beihang University, Beijing 100000, China; sy1902106@buaa.edu.cn (S.Z.); rongke_liu@buaa.edu.cn (R.L.); zhaoweiting@buaa.edu.cn (W.Z.)

**Keywords:** 5G broadcasting, mmWave, tunnel, SIW horn antenna, MOGA

## Abstract

5G mmWave broadcasting is considered to be the main method of future broadcasting. However, the mmWave transmission has a large space loss, especially in tunnels. In order to compensate for the attenuation of the broadcast signal, a directional horn antenna is designed in this paper. Substrate integrated waveguide (SIW) technology, dual-element arrays and extension structures are used to improve the antenna structure and achieve good directional characteristics. The multi-objective genetic algorithm (MOGA) is used to optimize the antenna parameters and improve optimization efficiency. Ultimately, the whole antenna was 28.2 mm in length and 28.6 mm in width, and the FR4 material, with a relative permittivity of 4.4, was used as a dielectric plate. The maximum gain of the antenna is 8.06 dB, and the bandwidth with gain greater than 6.5 dB is nearly 2 GHz. Antenna performance simulation and test results show that the extended semicircular structure is beneficial in enhancing the directional radiation of the antenna. This provides a reference method for directional antennas applied to 5G millimeter wave bands to increase gain and narrow beams.

## 1. Introduction

With the commercial deployment of fifth generation (5G) mobile networks, broadcasting services have become diversified [1,2,3]. The integration of traditional TV services and emerging media, such as internet protocol television (IPTV) and short video services, has become a new trend in 5G broadcasting services [4]. In the future, the 5G broadcasting services will develop in the direction of 4 K/8 K ultra-high-definition (Ultra HD), smooth video, and immersive virtual reality (VR) [5,6,7]. However, these high-quality services pose huge challenges in allocating limited spectrum resources in 5G broadcasting.

Recently, the millimeter wave (mmWave) band with wide bandwidth has become an important research direction for 5G broadcasting [8,9,10]. According to the 5G frequency band specified by 3GPP, the 5G mmWave band range is 24–52 GHz. Compared with the low frequency band, it has a larger bandwidth and generates less reflection loss and less refraction loss [11]. In addition, the successful application of 5G mmWave channels in TV services proves the broad prospects of mmWave in 5G broadcasting [12]. From the above, mmWave broadcasting is an inevitable trend of future broadcasting development. However, there are two main problems with mmWave transmission in 5G broadcasting: (1) The mmWave signals with short wavelengths are easily interfered by external noise and attenuated to varying degrees. Especially in tunnels, tunnel materials with high dielectric constant or conductivity will accelerate transmission loss. (2) The curved shape of the tunnel will cause more losses in the tunnel. The path loss of curved tunnels is much greater than that of straight tunnels [13]. The greater the degree of curvature of the tunnel, the greater the delay spread and the lower the received power [14]. To solve this problem, the method of adopting mmWave directional antennas for 5G broadcasting in tunnels has been proposed.

Installing a directional antenna with high gain in the tunnel is an effective method to resist propagation attenuation [15]. Moreover, if the tunnel wall is smooth, the path loss will be alleviated [16]. The multi-input-multiple-output (MIMO) antenna is a typical example of increasing gain and beam controllability in the form of a multi-element array [17,18,19], and some methods of enhancing gain by using special feed structures are also presented [20]. Removing coupling between large array elements has also become a research hotspot [21]. However, multi-port input and multi-port output are extremely complex. The horn antenna, which is an important branch of the directional antennas, could achieve good gain performance and its structure is rather simple [22,23]. In the actual tunnel measurement process, a rotatable and controllable high-gain horn antenna is used to avoid loss [24]. The horn antennas have two traditional forms: (1) waveguide and (2) microstrip. Conforming to the current trend of antenna miniaturization, the substrate integrated waveguide (SIW) is proposed as a new transmission form of horn antennas [25]. The SIW not only has the advantages of large quality factor and small radiation loss of the waveguide, but also has the advantages of small dimensions, light weight and easy integration of the microstrip. Multitudes of performance improvement methods based on the SIW horn form have been proposed. The method of inserting some nails vertically into the antenna plane at the horn opening is presented to increase the antenna gain. The additional reflective nails enable the SIW horn antenna to obtain a gain of 13.97 dBi at 20.5 GHz [26,27,28]; nonetheless, immense amounts of reflector nails increase the difficulty of antenna manufacturing. Therefore, the array form of SIW horn antennas is proposed to increase the gain and narrow beam, which expands the coverage distance of 5G broadcast signals.

The major contributions of this paper include: (1) a mmWave horn antenna that meets the requirements of directional antennas in 5G broadcasting is designed; (2) a novel structural design method is proposed to improve radiation directivity; and (3) the multi-objective genetic algorithm (MOGA) is used to optimize antenna parameters and save optimization time. Through detailed performance evaluations, this study shows that the proposed mmWave directional antenna can meet the expectations of 5G broadcasting services.

The rest of the paper is organized as follows. We first introduce the prospects of 5G mmWave broadcasting in Section 2. We then propose our design approach for the mmWave horn antenna to reduce the propagation delay of 5G broadcasting services in Section 3. In Section 4, we present the numerical results to demonstrate the advantage of our proposed approach. In Section 5, we conclude this work.

## 2. System Model

Converging mobile broadband and terrestrial broadcast systems globally will enable mobile broadband operators to transfer a large portion of data traffic to broadcast networks, including many of the most popular video services that multiple users may need simultaneously. Broadcasters can benefit from the return channel of broadband networks to provide interactive on-demand services. In this section, we introduce a 5G broadcast coverage system and investigate the antenna demand for 5G broadcasting.

### 2.1. Broadcast Coverage System

With the rapid advance of IoT, the demand for high-speed network services on cruise ships and high-speed rail systems has surged. At present, the large population of the world makes land resources scarce, and underground passages and tracks are constantly emerging. Large-scale construction of highways, railways, tunnels, urban underground passages and subsea tunnels has become an inevitable trend. In diverse tunnels, the length of the tunnel varies from hundreds of meters to tens of kilometers. In order to ensure the safety of traffic in the tunnel and eliminate people’s concerns, the broadcast coverage in the tunnel is everyone’s expectation. Universally, tunnels are located under mountains or underground. Due to the randomness of the location of the transmitting station, the variability of the tunnel orientation, and the large bandwidth consumption of the fiber in the RF band, the fiber is not used as a long-distance transmission tool inside and outside the tunnel, but only used for short-distance transmission in the tunnel broadcast coverage system.

Tunnels of longer length are increasingly being constructed. The San Gotha baseline tunnel, the longest tunnel in the world, is 57 km long. Such a development trend is not conducive to the deployment of communication in the tunnel. On the one hand, the closed tunnel environment and the conductive material in the tunnel weaken the transmitted wave; on the other hand, the short-wave band accompanying high-frequency 5G will cause great energy loss to the mmWave transmission. Therefore, relay communication points must be deployed in long tunnels to transmit 5G mmWave with as little loss as possible. To achieve seamless signal coverage in underground tunnels, a similar cell design method to the base station (BS) distribution in a cellular network was used to configure relay station distribution [29]. In this paper, similar cellular network ideas are also presented to properly distribute repeaters to complete the broadcast signal coverage in the entire tunnel.

The transmission process, as illustrated in Figure 1, is produced. The broadcast signal is sent from the broadcast transmitting station, received in the air outside the tunnel, processed in the tunnel broadcast coverage system, retransmitted in the tunnel by the directional antenna, and eventually covered by the repeater. Due to the massive loss of mmWave, the signal must first undergo power amplification and other processing before it can be retransmitted in the tunnel. When an emergency occurs, emergency calls, mobile terminals, etc., trigger emergency broadcasts. The signal processing unit performs emergency broadcast insertion.

### 2.2. Antenna Configuration in the Tunnel

While the styles of the tunnels are diverse, the distribution of tunnels stay linear, not planar. Consequently, compared with the radiation beam of the omnidirectional antenna, the directional beam is more suitable for transmitting signals in the tunnel. Combined with the characteristics of the tunnel distribution, the tunnel length is divided into several small sections by adopting the idea of cell coverage. The repeater in the tunnel is equivalent to the base station in the cellular network. In addition to radio frequency (RF) devices with a signal amplification function, antennas capable of transmitting and receiving signals must also be installed in repeaters. Directional antennas can reduce energy loss and increase gain. A certain length of signal coverage can be achieved by installing a bidirectional antenna or two unidirectional antennas on the repeater. Nevertheless, the radiation intensity of an antenna is limited, and the one-time signal transmission at the tunnel port cannot cover the entire tunnel. Instead of BSs in the cellular network, we used multiple repeaters capable of bidirectional radiation to cover the full length in this work.

In recent years, bidirectional antennas have become increasingly popular in the wireless transmission technology of coal mines. However, bidirectional antennas have very strict index requirements in design and manufacturing. Furthermore, the implementation of bidirectional radiation is not mandatory for an antenna to be bidirectional; basic bidirectional radiation characteristics can be achieved by adjusting the angle of two unidirectional antennas. Moreover, the radiation intensity of a bidirectional antenna is lower than that of a unidirectional antenna when the input power is constant. Besides, array antennas can also generate bidirectional radiation beams [30], which require immense amounts of array elements, resulting in the large size of antennas. As a result, unidirectional antennas are the best choice for repeater antennas in tunnels.

Based on the idea of cell coverage, the repeater antenna arrangement in a linear tunnel is illustrated in Figure 2. The two unidirectional antennas on the repeater point to both sides of the tunnel to achieve overlapping signal coverage. When a user is in the middle of the two repeaters, the signal is switched. By taking this distribution, seamless broadcast coverage in the tunnel can be realized.

The demand for directional antennas has continuously increased, and at the same time, the design approaches have also been constantly innovated. For instance, a compact dual-frequency directional antenna for the application of RFID readers has been proposed in [31]. Additionally, to obtain a high gain, a large slot antenna array has been designed [32]. Moreover, to avoid feeding networks of antenna arrays, single high-directive antenna elements are proposed [33,34]. Unexpectedly, the popularity of simplified structures has resulted in SIW technology being widely applied to antenna design. Multitudes of research indicate that the innovative structure at the opening of the SIW horn antenna can enhance the radiation directivity. The dielectric arc lens at the horn opening was proposed to improve the directional gain [35]. Subsequently, instead of dielectric arc lenses, the extended dielectric load at the planar SIW horn opening was investigated [36].

The previous research demonstrates convincingly that the transition structure of extended semicircular dielectric plates can enhance directivity. In the present study, the directional radiation intensity is further enhanced by extending the metal plate. Based on the concept that the array can increase the gain, an SIW array antenna with two antenna elements is designed to increase the directional gain. According to multiple performance indicators of the antenna, the MOGA is adopted to optimize the antenna parameters in the design process.

## 3. Antenna Structure and MOGA Optimization

### 3.1. SIW Technology

This section introduces the basic horn antenna form based on SIW technology and relevant theoretical analysis.

The simplest SIW horn antenna is formed by a horn shape enclosed by metalized through-holes on rectangular waveguide forms, which are represented by small circles in Figure 3. The antenna is an H-plane horn antenna, which has an open horn on the H plane (horizontal plane), and periodic metalized holes are used to replace the metal waveguide sidewalls of the traditional H-plane horn antenna. The upper and lower sides of the dielectric plate are plated with copper, which has the same function as the upper and lower metal walls of the traditional horn antenna waveguide. The structure of the two waveguides is illustrated in Figure 3. Moreover, these two waveguides can be replaced with each other when the distance between the adjacent metalized holes is sufficiently small.

There exists a solid theoretical basis for the conversion between the two waveguides. Regarding the rectangular metal waveguide and its corresponding SIW, the conversion relationship is given in [37]. A coordinate axis is established in the conventional rectangular waveguide, as presented in Figure 4. It is assumed that the TE_10_ mode wave propagates inside.

The electromagnetic component of the rectangular waveguide is shown in the following equations.
(1)$$H_{z}=A\sin\frac{\pi x}{a}e^{-j\beta z}$$
(2)$$E_{y}=\frac{j\omega \mu }{\pi a}A\cos\frac{\pi x}{a}e^{-j\beta z}$$
a is the width of the rectangular metal waveguide, ω is the angular frequency, μ is the permeability, β is the propagation constant, and A is a constant. The longitudinal impedance of the waveguide is defined as
(3)$$\eta_{s}=\frac{E_{y}}{H_{z}}=\frac{j\omega \mu_{0}}{\left(\pi /a\right)}\cot\left(\frac{\pi }{a}x\right).$$
$\mu_{0}$ is the vacuum permeability. The surface impedance of the SIW is unrelated to the incidence angle of the wave. If the characteristic of the rectangular metal waveguide is the same as that of the SIW, the following equation can be obtained: (4)$$\frac{j\omega \mu_{0}W}{4}\ln\left(\frac{W}{2D}\right)=\frac{j\omega \mu_{0}}{\left(\pi /a\right)}\cot\left(\frac{\pi }{a}\frac{a\prime }{2}\right),$$
where a′ is the width of the SIW, W is the distance between two adjacent metalized holes, a is the width of a conventional rectangular waveguide, and D is the radius of the metalized hole.

A formula for the conversion between the SIW waveguide and the conventional rectangular waveguide is given as follows:(5)$$a\prime =\frac{2a}{\pi }ctg^{-1}(\frac{\pi W}{4a}\ln\frac{W}{2D}),$$

Substitution of Equation (5) into Equations (1) and (2) yields the relationship between the SIW structure and the radiation field.

However, there are some differences between the design of the SIW and that of traditional waveguides. To ensure that periodic metallized holes have a small loss, the metallized hole must meet the following requirements [38]:(6)$$D<\frac{\lambda_{g}}{5},$$
(7)W<2D,
where $\lambda_{g}$ is the wavelength propagating in the SIW.

### 3.2. Array Form and Semicircular Extension Structure

This section introduces the antenna structure according to its structural characteristics. Particular focus is given to the array form and extension structure.

There is no doubt that arrays help to improve antenna gain, but too many array elements will increase the complexity of the antenna. Therefore, the designed antenna is a dual-element array antenna. The H-plane beamwidth of the antenna is improved by merging two simple SIW horn antennas into an array antenna. The radiation intensity of SIW array antenna is doubled in the array form. This approach improves the beamwidth of the H-plane antenna at the expense of increasing the beamwidth of the E-plane of the antenna, which has little effect on the radiation characteristics.

There exist two considerable aspects in the array design process. One is the combination of two elements. The array antenna has a sole input, and thus, a transition between the main waveguide and the two waveguide elements is adopted. There appear four common curve structures: the right-angle curve structure, the straight-cut curve structure, the smooth curve structure, and the diagonal curve structure, as shown in Figure 5. Compared with the other three structures, the smooth curve structure not only optimizes the bandwidth, but also reduces the reflection coefficient, and consequently, it is the best choice for the transition structure [39].

Moreover, an additional metalized hole is added to the center axis of the antenna to ensure complete wave transmission at the shunt. The hole is assumed to be a lateral distance of b and a longitudinal distance of c from the end point of the main waveguide. The width of the main waveguide is denoted by a′. When their relationship satisfies Equation (6), the transmission loss between the standing waveguide and the secondary waveguide will be reduced.
(8)$$\sqrt{b^{2}+c^{2}}\approx a\prime$$

According to the optimization of antenna element, a′ is 5.2 mm, and b is 2.6 mm. According to Equation (6), when c is 4.5 mm, the wave is transmitted with low loss, as shown in Figure 6.

The other is the distance between the two elements. The array antenna consists of two horn antennas, and the distance between the two horn antennas has an effect on the antenna gain and beamwidth. When the open ends of two horn elements share the same metalized hole, the distance between the two horn antennas is zero. Different distances cause different performances of the array antenna, so the setting of the distance value is significant. 

For the surface antenna, the gain is usually expressed as follows. It can be found that the gain is directly proportional to the actual aperture area. Therefore, extending the antenna structure at the antenna opening can theoretically increase the gain.
(9)$$G=\frac{4\pi A_{a}}{\lambda^{2}}\eta ,$$
where G is the antenna gain, $A_{a}$ is the actual area of the aperture, and λ is the working wavelength. η refers to the antenna efficiency, which is generally expressed as
(10)$$\eta =\eta_{r}\eta_{c}\eta_{d}.$$
$\eta_{r}$ is the reflection efficiency, $\eta_{c}$ is the conduction efficiency, and $\eta_{d}$ is the dielectric efficiency. Increasing the aperture area $A_{a}$ is a feasible method to improve the gain. The semicircular extension structure is one of the characteristics of the antenna structure. Improving the radiation performance of the antenna has consistently been the focus of attention in related research, and the addition of a dielectric guide structure at the opening of the horn antenna is a simple and effective improvement. Consequently, the antenna will be characterized by a wide bandwidth, high efficiency, and low noise. This approach is based on the principle of internal reflection and negates the trade-off between the taper and overflow power in the traditional concept, which is a principle that is universally used in optics [35]. The function of this structure is very similar to that of a lens, and the structure seems to be an extension of a conventional parabolic lens. The concentration of the radiation beam in the axial direction of the antenna is enhanced by extending the length of the structure at the horn opening [36]. Utilizing this concept, each horn element in the array antenna is designed with a semicircular extended dielectric plate at the opening. The radius of the extended semicircular dielectric plate has a substantial influence on the gain of the antenna and must be optimized. However, its optimization range has certain limitations; the thickness of the dielectric plate must be less than one-sixth of the wavelength, otherwise, the efficiency of this method will be reduced [40].

Similar to extending the semicircular dielectric plate, the semicircular metal sheet is extended to further extend the axial length of the antenna. This provides a complete transition from the antenna to free space, which mitigates the loss of energy at the horn opening. After expanding the structure twice, the antenna bandwidth is further increased. The top view of the antenna after extending the semicircular metal sheet is presented in Figure 7.

After the expansion structure is completed, the final structure of the antenna is also obtained. An additional issue worthy of further discussion is that of feeding. The common feeding methods of SIW horn antennas are metal waveguide feeding and coaxial line feeding. Compared with coaxial line feeding, metal waveguide feeding is characterized by a simple structure; however, the thickness of the dielectric plate of the metal waveguide is small, which will lead to low feeding efficiency. Coaxial feeding has a large power capacity when the impedance of the antenna matches the impedance of the transmission line, thereby resulting in a high gain, large efficiency, and small return loss of the antenna.

### 3.3. MOGA Optimization Scheme

The genetic algorithm (GA) is a search method based on the natural evolution to solve optimization problems. It takes all individuals in the group as the object and uses randomization technology to search for the global optimal solution. However, if the parameter setting is not reasonable, the optimal solution cannot be obtained. Compared with the traditional optimization method, this algorithm is characterized by high accuracy, high efficiency, and high flexibility [41]. Therefore, the GA is the best choice to obtain the optimal solution for the sole objective.

In practical problems, there are usually multiple objectives. If GA is used to optimize the target one by one, the optimal solution can be obtained for each objective. However, the relationships between multitudes of objective functions are not necessarily mutual promotion; they may produce the opposite effect, and the optimal value can sometimes not be reached. Furthermore, solving the objective one by one increases the time in proportion to the number of objective functions. In the past, the method of weighting each objective and then solving a single objective is usually used to solve multi-objective optimization problems. MOGA optimizes multiple objectives at the same time, which not only reduces waiting time but also simplifies computational complexity. Non-dominated sorting genetic algorithm-II (NSGA-II) is a typical method based on MOGA. The complexity of NSGA-II is represented by $O(mN^{2})$, where m is the number of targets and N is the population size. The weighting method reduces m, but greatly increases the power exponent term. NSGA-II is used to optimize the antenna structure; it mainly realizes multi-objective optimization from three aspects, namely a fast non-dominated sorting algorithm, an individual crowding distance operator and an elite strategy design. The execution process is presented in Figure 8.

The fast non-dominant sorting algorithm makes the search area move towards the Pareto optimal solution set by stratifying the population. Individuals within the same layer have the same non-dominated sorting value, $i_{rank}$, which is the non-dominated sorting variable. After that, the individual crowding distance is proposed to distinguish good and bad individuals with the same $i_{rank}$ value. The crowding distance of individual *i* is the distance between the two adjacent individuals, *i* + 1 and *i* − 1. The initial distance of individuals in the same layer is set to
(11)$$L{\left[i\right]}_{d}=0,$$
where $L{\left[i\right]}_{d}$ is the value of the crowding variable for individual *i*. Individuals at the sorting edge have a selection advantage. Their crowded distance is expressed as
(12)$$L{\left[1\right]}_{d}=L{\left[N\right]}_{d}=\infty .$$

For individuals whose sorting is not marginal, the calculation process is as follows:(13)$$L{\left[i\right]}_{d}=L{\left[i\right]}_{d}+(L{\left[i+1\right]}_{m}-L{\left[i-1\right]}_{m})/(f_{m}^{\max}-f_{m}^{\min}),$$
where $L{\left[i+1\right]}_{m}$ is the *m*-th objective function value of the (*i* + 1)-th individual, and $f_{m}^{\max}$ and $f_{m}^{\min}$ are the maximum and minimum values of the *m*-th objective function value, respectively.

For a specific objective function, by sorting the individuals in the same layer, the individuals with larger crowding distances are selected preferentially, which increases the uniformity of the distribution of the calculated results and is beneficial in finding the optimal solution. After crossover and mutation, the elite strategy is used to directly retain the good individuals of the parents to the offspring in order to prevent the loss of the Pareto optimal solution. Aiming at the two objective functions, the population, $R_{i}$, synthesized by $P_{i}$ and $Q_{i}$, was optimized through the principle of preferentially selecting the low $i_{rank}$ layer and the large crowding distance to form the new parent, $P_{i+1}$.

### 3.4. Optimization Parameters

In order to derive the horn antenna’s gain, directionality needs to be derived first, which means finding the maximum radiation. It is assumed that the field produced by the antenna under normal conditions is $E_{0}$. Then, the maximum electric field in spherical coordinates can be expressed as
(14)$$\begin{array}{c} {\left|E\right|}_{\max}=\sqrt{{\left|E_{\theta }\right|}_{\max}^{2}+{\left|E_{\phi }\right|}_{\max}^{2}}=\left|E_{0}\right|\frac{H}{r}\sqrt{\frac{L}{2\lambda }}{\left\{{\left[C(u)-C(v)\right]}^{2}+{\left[S(u)-S(v)\right]}^{2}\right\}}^{\frac{1}{2}}\\ \left\{\begin{array}{c} {\left|E_{\theta }\right|}_{\max}=\left|E_{0}\right|\frac{H}{r}\sqrt{\frac{L}{2\lambda }}\left|\sin\phi \{[C(u)-C(v)]\}-j[S(u)-S(v)]\right|\\ {\left|E_{\phi }\right|}_{\max}=\left|E_{0}\right|\frac{H}{r}\sqrt{\frac{L}{2\lambda }}\left|\cos\phi \{[C(u)-C(v)]+j[S(u)-S(v)]\}\right|\\ u=\frac{1}{\sqrt{2}}\left(\frac{\sqrt{\lambda L}}{A}+\frac{A}{\sqrt{\lambda L}}\right)\\ v=\frac{1}{\sqrt{2}}\left(\frac{\sqrt{\lambda L}}{A}-\frac{A}{\sqrt{\lambda L}}\right), \end{array}\right. \end{array}$$
where $C\left(u\right)$ and $S\left(u\right)$ are the Fresnel cosine integral and sine integral, respectively, H is the thickness of dielectric plate, and A is the horizontal length of the horn. If θ is the horn opening angle, arctan $\left(\theta /2\right)=A/2L$.

The directivity coefficient can be calculated by the following equation:(15)$$\begin{array}{l} D=\frac{4\pi U_{\max}}{P_{rad}}\\ \left\{\begin{array}{c} U_{\max}=\frac{r^{2}}{2\eta }{\left|E\right|}_{\max}^{2}\\ P_{rad}={\left|E_{0}\right|}^{2}\frac{HA}{4\eta } \end{array}\right. \end{array}$$

The antenna gain can be expressed as follows [42]:(16)$$\left\{\begin{array}{c} G=D\eta =\eta \frac{4\pi HL}{\lambda A}\left\{{\left[C(u)-C(v)\right]}^{2}+{\left[S(u)-S(v)\right]}^{2}\right\}\\ u=\frac{1}{\sqrt{2}}\left(\frac{\sqrt{\lambda L}}{A}+\frac{A}{\sqrt{\lambda L}}\right)\\ v=\frac{1}{\sqrt{2}}\left(\frac{\sqrt{\lambda L}}{A}-\frac{A}{\sqrt{\lambda L}}\right) \end{array}\right.$$

The size of the antenna has a great influence on its radiation performance, and as a result, it is an important parameter to optimize. According to the application requirements of the antenna, the gain and return loss of the antenna are taken as the objective optimization functions. In the simple SIW horn antenna design process, the three variables, H, A, and the SIW width a′, are optimized as optimization variables. In the design of the SIW array antenna with an improved structure, the distance, W, between the two horns, the radius, R, of the extended semicircular dielectric plate, and the radius, r, of the extended semicircular metal sheet are used as optimization variables to optimize the antenna performance. Both antenna optimization problems include two objective functions and three optimization variables.

The size of the millimeter level is very small, but the resulting performance difference is large. Therefore, the parameter optimization range and optimization times are small. The number of optimizations is limited to 200, and the number of individuals in the population is 20. The cell antenna and array antenna are optimized separately, and the MOGA is executed twice. However, in both cases, the optimization variables are different but the objective function is the same.

The multi-objective optimization model can be expressed as
(17)$$Optimal\;\bar{F}(\bar{x})=[F_{1}(\bar{x}),F_{2}(\bar{x})],$$
where $\bar{F}(\bar{x})$ is the vector of the objective function composed of two objective functions $\bar{F_{1}}(\bar{x})$ and $\bar{F_{2}}(\bar{x})$.
$\bar{x}=[\bar{x_{1}},\bar{x_{2}},\bar{x_{3}}]$, $\bar{x_{i}}=[x_{i1},x_{i2}\ldots ,x_{iN}]$, and N=20. $\bar{x}$ is a vector composed of three independent variables. For a simple SIW horn antenna, $\bar{x_{1}}\in [2.5,3.5],$
$\bar{x_{2}}\in [0.2,0.48],$ and $\bar{x_{3}}\in [4.2,5.8]$ (unit: mm; the $\bar{x_{1}}$ is H, the $\bar{x_{2}}$ is A that represents the horn opening angle, and the $\bar{x_{3}}$ is a′). For an SIW array antenna, the $\bar{x_{1}}$ is W, the $\bar{x_{2}}$ is R, and the $\bar{x_{3}}$ is r. Therefore, $\bar{x_{1}}\in [0,1.8],$
$\bar{x_{2}}\in [5,11],$ and $\bar{x_{3}}\in [0.5,5.5]$ (unit: mm).

To ensure sufficient bandwidth, two frequency sampling points were selected in the calculation model; namely $f_{1}=25GHz$ and $f_{2}=26GH$. There are certain setting requirements for the two objective functions. Return loss is an important factor by which to judge the matching of a connection system in various circuits and communication fields. Normally, the value of $S_{11}$ is required to be less than −15 dB. Return loss is generally required at
(18)$$S_{11}(f_{i})=\{\begin{array}{c} S_{11},S_{11}<-15dB\\ -15dB,S_{11}\geq -15dB \end{array},$$
(19)$$F_{1}(\bar{x})=\frac{1}{N}\sum_{i=1}^{N}S_{11}^{}(f_{i}).$$

Antenna gain is a measure of an antenna’s ability to transmit and receive signals in a particular direction. It is one of the most important parameters by which to measure the performance of a directional antenna, the average gain value of which is required to be greater than 6.5 dB. The objective function of gain is the average gain at the working frequency.
(20)$$F_{2}(\bar{x})=\frac{1}{N}\sum_{i=1}^{N}Gain(f_{i})$$

Therefore, when the return loss is less than −15 dB and the average gain value is greater than 6.5 dB, the multi-objective function can be achieved.

## 4. Simulation and Test Results

In this study, HFSS based on the finite element method is used for antenna design. There exist some default parameters settings as follows. The FR4 material with a relative permittivity of 4.4 and a dielectric loss tangent of 0.02 was used as a dielectric plate, and a coaxial line with a characteristic impedance of 50 Ω was used for the antenna feeding. The whole antenna was 28.2 mm in length and 28.6 mm in width. The diameter of the metalized cylindrical hole was 0.4 mm, and the distance between the two adjacent holes was 1 mm. The radius of the coaxial line was 0.43 mm, and the radius of the excitation port was 0.88 mm. The distance between the center of the coaxial line and the boundary of the dielectric plate was 4.5 mm. 24 GHz–30 GHz is the most popular research frequency range in the 5G mmWave band; in this work, antennas were primarily studied at 26 GHz.

### 4.1. HFSS Simulation Results

The parameter optimization was divided into two steps. First, the parameters of the simple SIW horn antenna were optimized by MOGA. Then, the dimension parameters of the improved SIW array antenna were optimized by MOGA. The objective function of the antenna is presented in Table 1, and the final optimization values of the parameters are reported in Table 2.

The improved antenna structure was compared with the simple SIW array antenna structure, as presented in Figure 9. The performance of the antenna was simulated at 26 GHz, and the performance of the improved antenna was compared with that of the simple SIW array antenna to obtain the H-plane radiation pattern, as exhibited in Figure 10.

The simple SIW array antenna had a half-power beamwidth of 34 degrees and a maximum gain of 2.70 dB, whereas the improved antenna had a half-power width of 28 degrees, a maximum gain of 8.06 dB, and a sidelobe level of −4.23 dB. The beamwidth was optimized by 6 degrees and the maximum gain was increased by 5.36 dB, which indicated the effectiveness of the extended semicircular structure for antenna radiation.

### 4.2. Anechoic Chamber Test

Four different antennas were fabricated to verify the effectiveness of the improved structure. These four antennas were, respectively, labeled as antenna one, antenna two, antenna three and antenna four. Antenna one was the final antenna optimized during the simulation, and its structure was similar to that illustrated in Figure 7. The difference between antennas one and two is that the W in antenna two was 2 mm greater than that in antenna one. The difference between the antennas three and one is that the R in antenna three was 5 mm smaller than that in antenna one. The difference between the antennas four and one is that antenna four did not have a semicircular extended metal sheet at the horn opening. The specific parameter values of the four antennas are reported in Table 3, and their structures are depicted in Figure 11. The antennas were manufactured in accordance with the designed structural parameters. Production machines, however, may cause minor errors. 

The test results of antenna two were compared with those of antenna one. The difference between the maximum gain verifies the effectiveness of optimizing W to improve the radiation directivity and gain. By comparing antenna three with antenna one, it can be confirmed that the R affected the maximum gain of the antenna. By comparing antenna four with antenna one, it can be discerned that the expanded metal sheet structure optimized the radiation beamwidth. The test results are presented in Figure 12. After comparing these results, the effect of antenna aperture size on radiation performance can be clearly observed, and the directional gain decreases when the antenna aperture is reduced.

The comparison of the measured and simulation performances of antenna one is presented in Figure 13, in which the measured radiation curve is approximately the same as the simulated radiation curve. However, the measured maximum gain was 7.01 dB, whereas the simulated maximum gain was 8.06 dB. There are three main reasons for this difference. First, loose contact between the coaxial cable and the dielectric plate can cause energy leakage. Second, there were inaccuracies in the manufacturing process. Finally, the accuracy of the test equipment was not sufficient. These factors led to deviations in the test results and gain loss. According to the measured reflection coefficient curve shown in Figure 14, the antenna operated over a wide range of frequencies, and the return loss was −30.89 dB at 25.34 GHz, which proved that the antenna can achieve a perfect match. In the same range, the change curves of gain and efficiency with frequency are presented in Figure 15. The curve of antenna efficiency with respect to frequency is almost identical to the curve of gain with respect to frequency. The gain was consistent with the required value in the range of 25–27 GHz. Moreover, the gain in this range was significantly improved compared to the gain at 24 GHz. The bandwidth of the antenna is approximately 2 GHz, using the optimization objective as the standard.

There are numerous design methods of the H-plane horn antenna. The antenna in [36] also adopts the method of array and extended dielectric plate. The large number of antenna elements brings strong radiation intensity and leads to a large antenna size at the same time. In addition to the difference of the extended metal sheet, the transition from the primary waveguide to the secondary waveguide in [36] uses a right-angled curve structure and the antenna has two input ports, resulting in an undesirable S parameter and a bandwidth of around 0.5 GHz. The antenna in reference [43] has a completely different structure. Two linear rows of triangular metal sheets are printed at the antenna opening to enhance antenna matching. The antenna has the characteristics of miniaturization and wide bandwidth, but its directional beam and S parameter are not outstanding. Therefore, by comprehensive analysis of all parameters, the proposed structure has certain advantages in S parameters, bandwidth and radiation beam.

## 5. Conclusions

5G mmWave transmission suffers from large path loss, especially in narrow long tunnels. This paper, correspondingly, develops a directional antenna for 5G broadcasting signals to propagate well in tunnels. In particular, we designed a miniaturized horn directional antenna suitable for ultrahigh frequency. Our design employs SIW technology to integrate a horn antenna array, an expanded dielectric board and an expanded metal plate. We also utilized MOGA to optimize the antenna parameters and reduce the optimization time. Simulation and field test results reveal that our developed antenna exhibits an improved performance characterized by high gain, narrow beam and wide bandwidth.

The antenna is 28.2 mm × 28.6 mm in size and 3 mm thick. The distance between the array elements is 1.4 mm, the radius of the extended dielectric plate is 10 mm, and the radius of the metal sheet is 3 mm. The antenna has a maximum gain of 8.06 dB at 26 GHz and a bandwidth of nearly 2 GHz. The antenna was measured with perfect reflection coefficient in the 24–28 GHz range. The semicircular structure, similar to a parabolic lens, acts as a lens to condense the beam. In addition, the guidance structure provides a complete transition from the antenna to free space for the transmitting wave. However, there is a limitation to this structure: the thickness of the dielectric plate must be less than 1/6 of the wavelength, otherwise the efficiency of this method will be reduced.

## Figures and Tables

**Figure 1 sensors-21-00746-f001:**
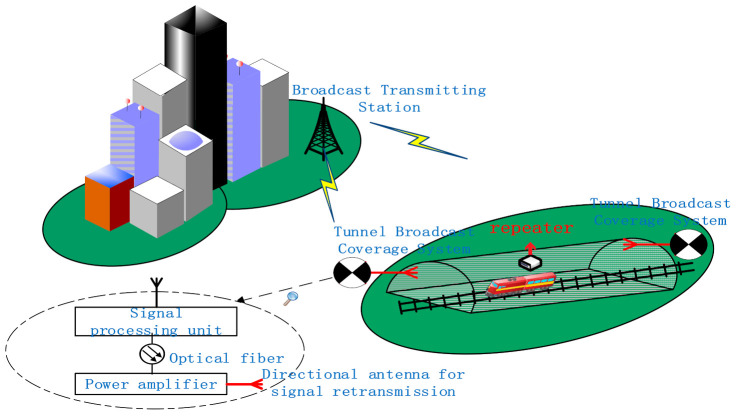
Schematic diagram of broadcast coverage system.

**Figure 2 sensors-21-00746-f002:**
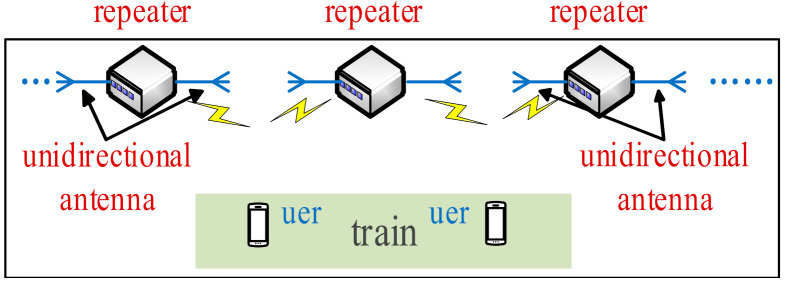
Antenna distribution.

**Figure 3 sensors-21-00746-f003:**
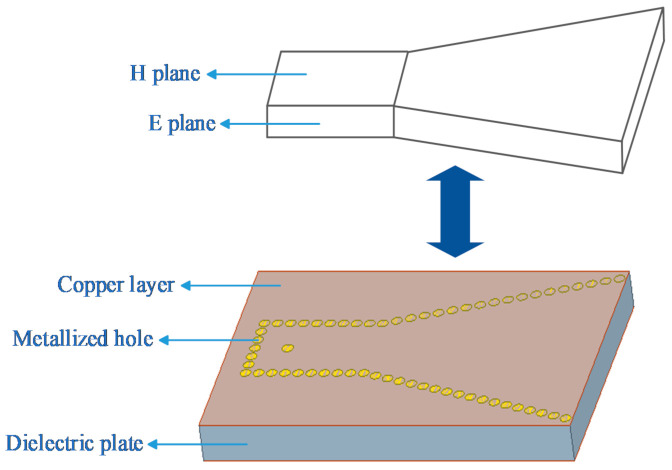
Traditional horn waveguide and simple SIW horn waveguide. (substrate integrated waveguide).

**Figure 4 sensors-21-00746-f004:**
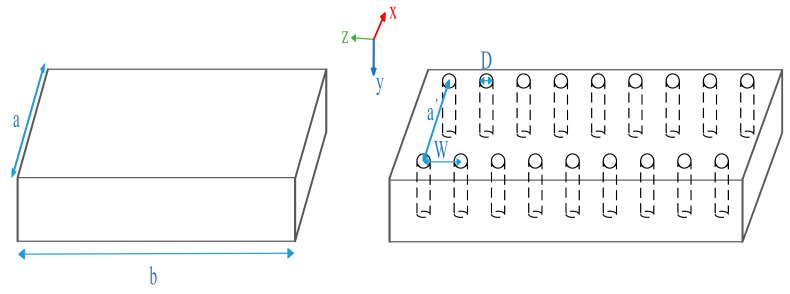
Traditional rectangular waveguide and simple SIW rectangular waveguide.

**Figure 5 sensors-21-00746-f005:**
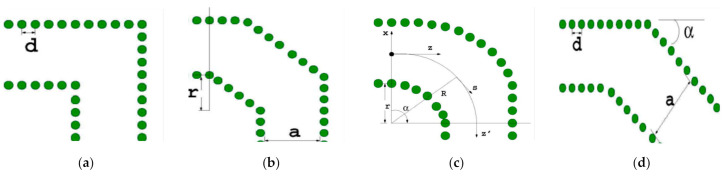
Different curve structures: (**a**) the right-angle curve structure; (**b**) the straight-cut curve structure; (**c**) the smooth curve structure; (**d**) the diagonal curve structure.

**Figure 6 sensors-21-00746-f006:**
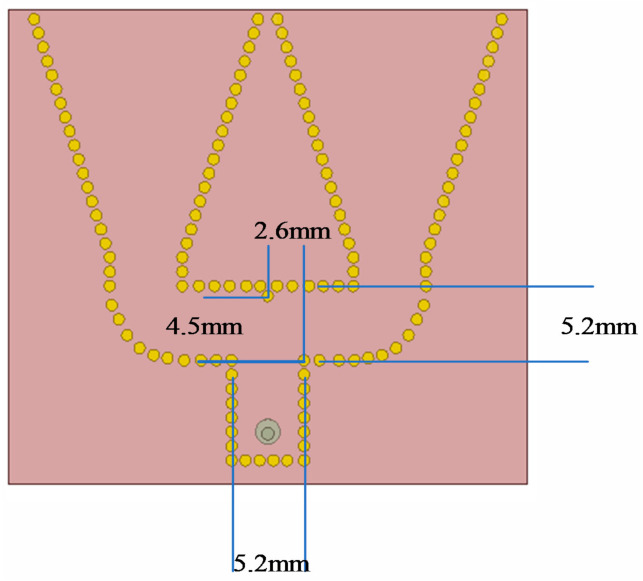
Location of the additional hole.

**Figure 7 sensors-21-00746-f007:**
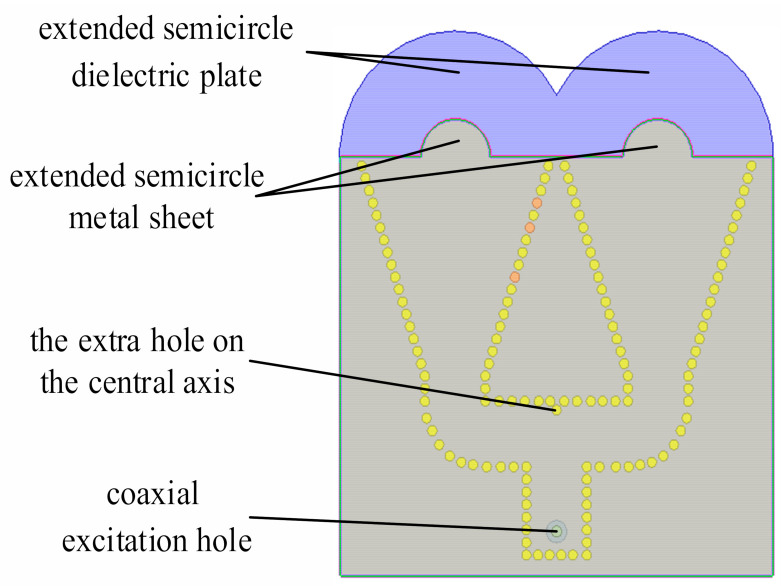
Schematic diagram of the improved antenna structure.

**Figure 8 sensors-21-00746-f008:**
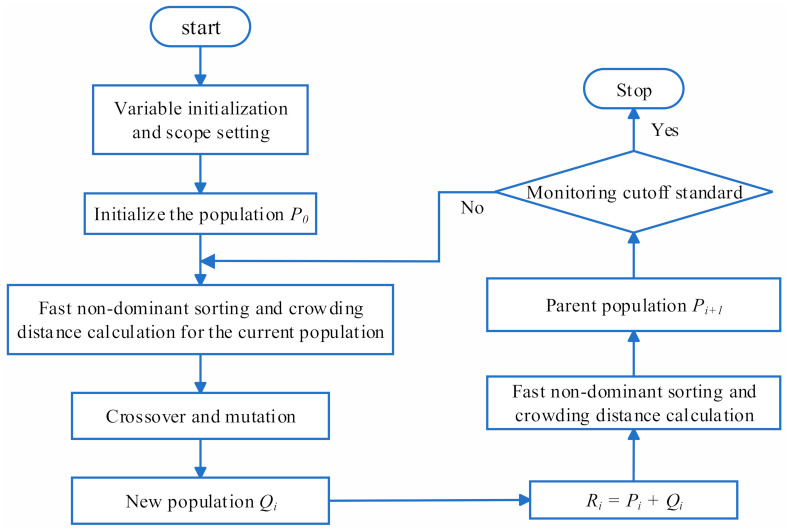
The flow chart of MOGA. (multi-objective genetic algorithm).

**Figure 9 sensors-21-00746-f009:**
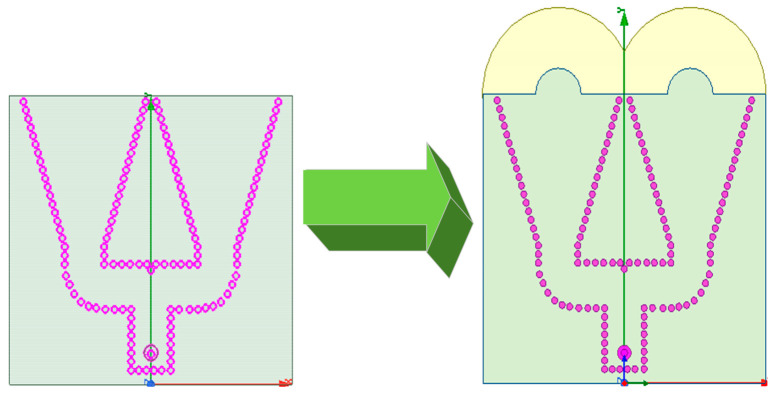
Comparison of antenna structure before and after improvement.

**Figure 10 sensors-21-00746-f010:**
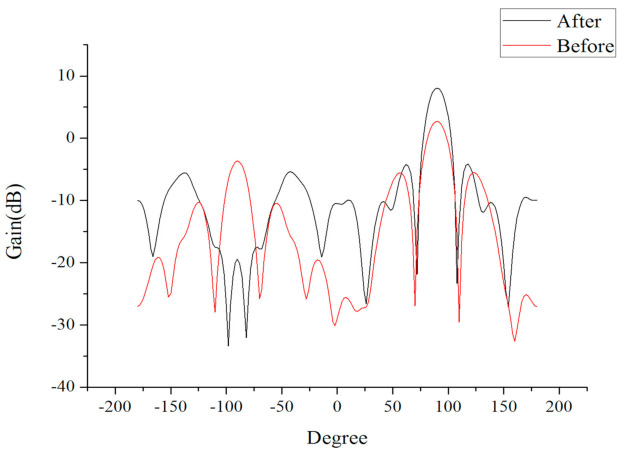
Performance comparison before and after improvement.

**Figure 11 sensors-21-00746-f011:**
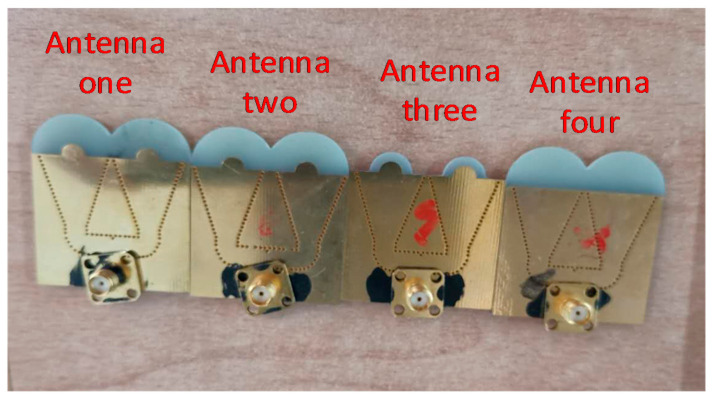
Comparison of four antennas.

**Figure 12 sensors-21-00746-f012:**
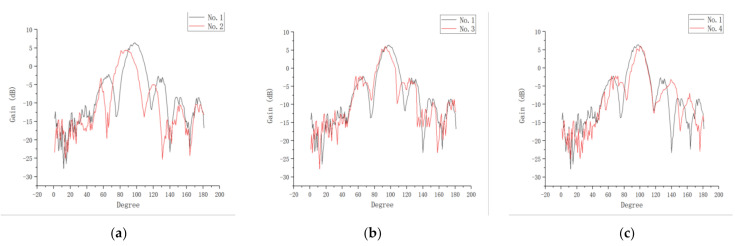
Measurements of the influence of the improved structure on antenna performance: (**a**) Antenna one and antenna two; (**b**) antenna one and antenna three; (**c**) antenna one and antenna four.

**Figure 13 sensors-21-00746-f013:**
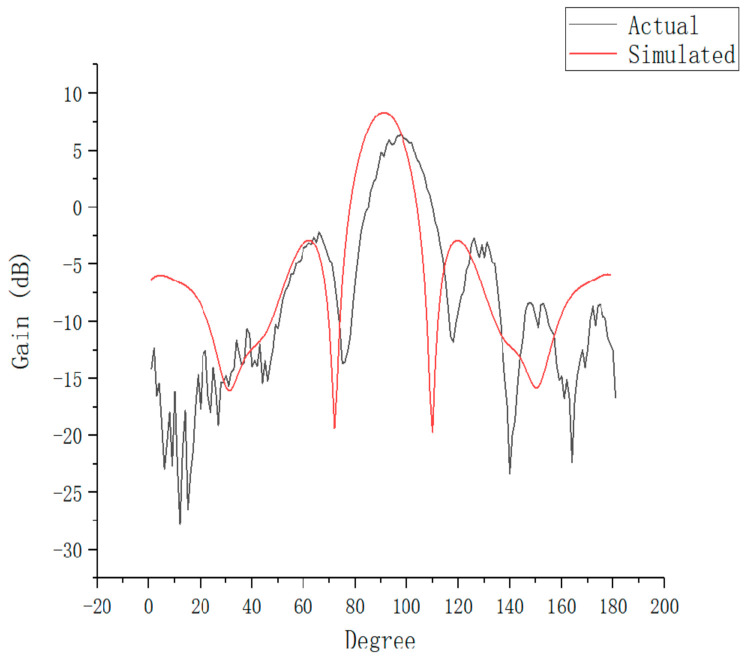
Comparison of test and simulation.

**Figure 14 sensors-21-00746-f014:**
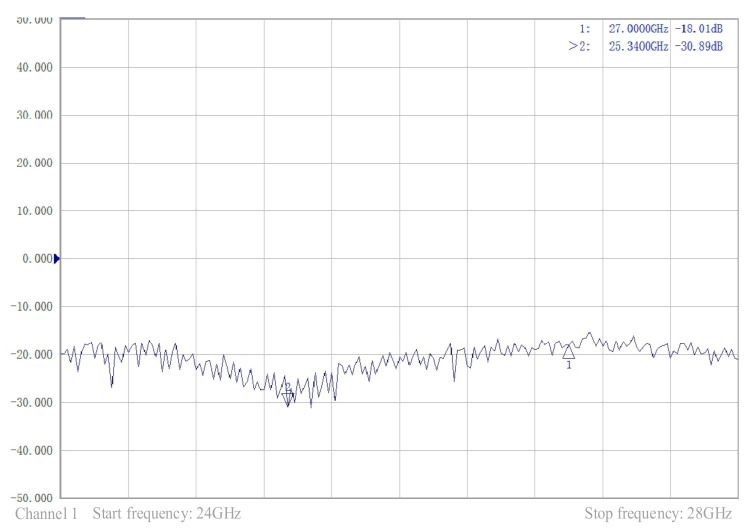
Reflection coefficient test.

**Figure 15 sensors-21-00746-f015:**
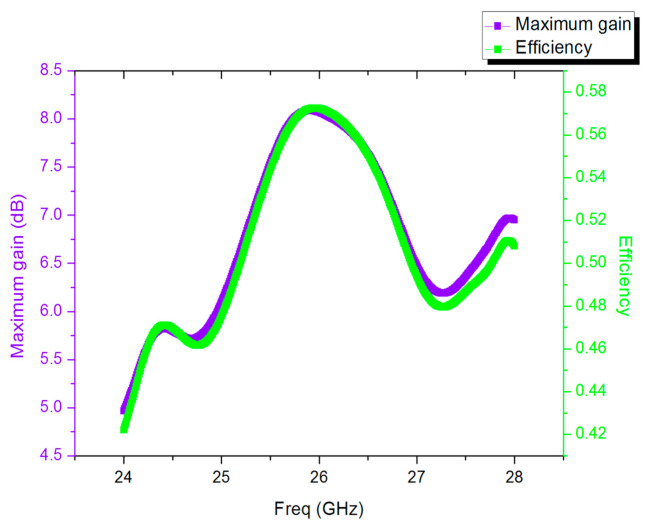
The change curves of gain and efficiency with frequency.

**Table 1 sensors-21-00746-t001:** Objective function values.

Index	Value
$S_{11}$	<−15 dB
Average antenna gain	>6.5 dB

**Table 2 sensors-21-00746-t002:** Parameter optimization table.

Antenna Type	Variables	Dimension (mm)
Single SIW horn antenna	H	3
	a′	5.2
	A	0.326
Improved antenna	W	1.4
	R	10
	r	3

**Table 3 sensors-21-00746-t003:** Parameter table.

Antenna	W (mm)	R (mm)	r (mm)
1	1.4	10	3
2	3.4	10	3
3	1.4	5	3
4	1.4	10	0

## Data Availability

The study did not report any data.

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
