# Peer review of "Design of mmWave Directional Antenna for Enhanced 5G Broadcasting Coverage"

_sensors, 2021, doi:10.3390/s21030746_

Round 1
Reviewer 1 Report
This paper designs a mmWave horn antenna to meet the requirements of directional antennas in 5G broadcasting and proposes a structural design method to improve radiation directivity. The topic is very interesting. The main contribution of this paper is to fabricate different antennas so that the proposed design and method can be actually verified. However, the following issues should be considered more clearly.
- The symbols used in equations must be explained. Too many symbols are not explained such that readers do not understand the equations and cannot follow the paper. Please explain all of the symbols.
- It is better to simplify some descriptions. For example, in Section II, there are too many descriptions. However, the system model is not complicated. Some useless sentences will really confuse readers.
- MOGA is proposed to reduce the optimization time. However, in Section 4, there is no comparison about MOGA. Does MOGA really reduce the optimization time? How long do other algorithms take? Are the parameters calculated by MOGA better than those of other algorithms?
Author Response
Thank you for your comments. I have uploaded the PDF file to answer your question.

Reviewer 2 Report
Dear Authors,
Here are my comments:
*) pag.5, after the sentence " a large slot antenna array has been proposed [27]"- >the following is suggested. "Also, to avoid feeding networks of antenna arrays, single high-directive antenna elements are proposed (ref)"
ref) J.Anguera, A. Andújar, J. Jayasinghe, “High Directivity Microstrip Patch Antennas Perturbing TModd-0 modes”, IEEE Antennas and Wireless Propagation Letters, vol.19, no. 1, pp. 39-43, 2020
ref) J. W. Jayasinghe, J. Anguera, and D. N. Uduwawala, "A High-Directivity Microstrip Patch Antenna Design by Using Genetic Algorithm Optimization", Progress In Electromagnetics Research C, Vol. 37, 131-144, 2013.
*) Figure 3. H plane, E plane should be E plane H plane
*) pag.6, lne 4. TE10, use subindex for 10
*) Pag.8, eq. (9), define after the equations letters. Explain for instance how illumination efficiency is computed for a SIW antenna
*) Figure 11. Enlarge, please
*) Can you comment on the fabrication process of the antennas?
*) Specify the er and loss tangent of the substrate
*) Improve the quality of figure 14
*) Can you show how gain changes versus the same frequency range shown in figure 14?
I hope these comments help
Thanks
Author Response

(The authors gave the same response as above.)

Reviewer 3 Report
Authors in this paper have proposed and manufactured a miniaturized horn directional antenna suitable for ultrahigh frequency. The idea and concept are interesting and the paper has been well organized. However, in this reviewer's point of view the authors are required to carefully address the following pints before final acceptance.
1) Abstract section needs to rewrite. Instead of providing a large background in this section, authors are suggested to provide discussions on the antenna's design process. Additionally, its performance parameters such as dimensions, bandwidth, radiation gain and efficiency should be mention in this part. Its novelty and practical applications should be highlight as well.
2) More antenna structures in the introduction section should be studied to provide a comprehensive background. In below there are proper suggestions.
"Integrated LTE and Millimeter-Wave 5G MIMO Antenna System for 4G/5G Wireless Terminals." Sensors 20, no. 14 (2020): 3926.
"A Comprehensive Survey of "Metamaterial Transmission-Line Based Antennas: Design, Challenges, and Applications"", IEEE Access, vol. 8, pp. 144778-144808, 2020.
"Electromagnetic bandgap backed millimeter-wave MIMO antenna for wearable applications." IEEE Access 7 (2019): 111135-111144.
"High-Gain On-Chip Antenna Design on Silicon Layer with Aperture Excitation for Terahertz Applications", IEEE Antennas and Wireless Propagation Letters, vol. 19, no. 9, pp. 1576-1580, Sept. 2020.
"Compact Millimeter-Wave MIMO Antenna for 5G Applications." In 2020 14th European Conference on Antennas and Propagation (EuCAP), pp. 1-5. IEEE, 2020.
"A Comprehensive Survey on "Various Decoupling Mechanisms with Focus on Metamaterial and Metasurface Principles Applicable to SAR and MIMO Antenna Systems"", IEEE Access, vol. 8, pp. 192965-193004, 2020, doi: 10.1109/ACCESS.2020.3032826.
3) Antenna structure in section 3 should be explain in more details. Please mention why the proposed layout has been selected in this design? Its advantages and disadvantages (if any) should be clearly highlight in section 3.
4) In Fig.3 authors can present the SIW with a different color to have better view.
5) The extracted derivations should be discuss in more details. How authors have extracted them?
6) It would be nice to add a comparison section before concluding the paper. In this section please compare the proposed work with prior arts in terms of dimensions, operational bandwidth, radiation gain and efficinecy.
7) In the manuscript besides the gain curves please add the efficiency curves over frequency as well.
8) Quality of the figures can be improve.
9) Reference part should b extend. In above there are appropriate suggestions.
10) Numerical results should be add in the conclusion section. Additionally, the advantages of the proposed antenna and its possible limitations can be add here.
Author Response

(The authors gave the same response as above.)

Reviewer 4 Report
Authors provide a good introduction and statement of the problem. Nevertheless, no indication is given regarding competing approaches and their performance. A comparison table should be provided, just before the conclusions, evidencing the relative position of the proposed antenna as compared to others (actually adopted or in open literature).
The MOGA that has been adopted for the optimisation seem a good approach, but the computational effort is not provided nor compared with othe possible ones. Please comment on the computational burden and compare.
What about the mutual interaction between the two antenna outputs ? please at least comment.
Please provide a bit more details for the experimental setup.
Fig. 3 mispells should be corrected.
what is (8) ? Using anumber only does not provide any insight. please provide the expression and then apply numbers ....
fig. 6 units are missing.
In general all the captions should be revised. Both in the text and captions please clearly indicate what is measured (do not use the word 'actual', since it may also refer to simulations.
Some mispells should be corrected all along the manuscript.
Author Response

(The authors gave the same response as above.)

Round 2
Reviewer 1 Report
All my comments have been addressed.
Reviewer 2 Report
Dear Authors,
Thanks for improving the paper.
I do have only one last comment. In Fig. 15, please move the legend inside the rectangular box of the plot.
Thanks